# Implications of kappa-casein evolutionary diversity for the self-assembly and aggregation of casein micelles

Jean Manguy[1,2,3] and Denis C. Shields[1,2,3]

[1]UCD Conway Institute, [2]School of Medicine, and [3]Food for Health Ireland, University College Dublin, Belfield, Dublin 4, Ireland

 JM, 0000-0002-8427-007X; DCS, 0000-0003-4015-2474

evolution/molecular biology/computational biology

casein, evolution, protein disorder, casein micelle

**Author for correspondence:**
Denis C. Shields
e-mail: denis.shields@ucd.ie

Milk alpha-, beta- and kappa-casein proteins assemble into casein micelles in breast epithelial cells. The glycomacropeptide (GMP) tails of kappa-casein that extend from the surface of the micelle are key to assembly and aggregation. Aggregation is triggered by stomach pepsin cleavage of GMP from para-kappa-casein (PKC). While one casein micelle model emphasizes the importance of hydrophobic interactions, another focuses on polar residues. We performed an evolutionary analysis of kappa-casein primary sequence and predicted features that potentially impact on protein interactions. We noted more rapid change in the earlier period (166 to 60 Ma). Pepsin and plasmin cleavage sites were avoided in the GMP, which may partly explain its amino acid composition. Short tandem repeats have led to modest expansions of PKC, and to large GMP expansions, suggesting the GMP is less length constrained. Amino acid compositional constraints were assessed across species. Polarity and hydrophobicity properties were insufficient to explain differences between PKC and GMP. Among polar residues, threonine dominates the GMP, compared to serine, probably reflecting its preference for O-glycosylation over phosphorylation. Glutamine, enriched in the bovine PQ-rich region, is not positionally conserved in other species. Among hydrophobic residues, isoleucine is clearly preferred over leucine in the GMP, and patches of hydrophobicity are not markedly positionally conserved. PKC tyrosine and charged residues showed stronger conservation of position, suggesting a role for pi-interactions, seen in other structurally dynamic protein membraneless assemblies. Independent acquisitions of cysteines are consistent with a trend of increasing stabilization of multimers by covalent disulphide bonds, over evolutionary time. In conclusion, kappa-casein compositional and positional constraints appear to be influenced by modification preferences, protease evasion and protein–protein interactions.

# 1. Introduction

Caseins are an important group of proteins in milk and are secreted by the mammary epithelial cells (MEC) [1], having evolved from the secretory calcium-binding phosphoprotein (SCPP) family of proteins [2]. Together these intrinsically disordered proteins self-assemble along with calcium phosphate (CaP) into casein micelles [3]. Casein micelles are a major source of amino acids, calcium and phosphates. Alpha-caseins and beta-caseins are mostly located inside the casein micelles and bind to the CaP while kappa-casein is mostly located on the surface [4,5]. In the mammary gland, the formation of casein micelles allows the concentration in liquid milk of protein and CaP [3,4]. Alone, caseins at high concentrations can form insoluble and toxic amyloid fibrils [6,7], and calcium at the concentration found in milk would precipitate [8]. In the neonates' stomachs, casein micelle aggregation coagulates milk.

Kappa-casein is critical to lactation [9]. It is indispensable for micelle formation, internal structure stabilization, avoidance of aggregation prior to ingestion, and the timely aggregation of the micelles in the stomach. Chymosin is a mammalian specialized newborn-specific gastric peptidase. In the stomach chymosin (CYM; EC 3.4.23.4) or pepsin (PGA; EC 3.4.23.1) cleaves kappa-casein, yielding two peptides: the glycomacropeptide (GMP) from the C-terminus and para-kappa-casein (PKC) from the N-terminus. Chymosin has a higher optimum pH than pepsin (5.8 and 2–5, respectively), a lower activity and a higher specificity for the PKC/GMP cleavage site than its paralogues, the pepsins. Functional chymosin was lost in different mammalian clades, including humans; in particular, it has been lost in clades where immunoglobulins may be transferred across via the placenta, perhaps because the need to avoid proteolytic damage to immunoglobulin-rich milk is less critical in these species [10,11]. Monotremes lost their acidic stomach so that no pepsin-like or chymosin-like activity is thought to be possible in their digestive system [12].

Prior to its cleavage, the negatively charged GMP region of kappa-casein is anchored to the micelle surface, thus repelling the aggregation of casein micelles in the mammary glands. The PKC region interacts with other kappa-casein chains and other caseins to stabilize the casein micelle's internal structure. Cysteines located in the PKC can covalently bond to form dimers, oligomers and in some species intrachain disulphide bonds [13]. Once the GMP is removed, the casein micelles aggregate together and form a gel [3]. This coagulating property is used to produce various dairy products. In addition to their digestion by gastric and intestinal peptidases, milk proteins can also be hydrolysed by peptidases already present in milk, in particular, plasmin [14]. The incomplete proteolysis of caseins releases peptides, some of which may exert biological activities on the organism and its gut microbiota [15,16].

Changes in pH are key to the functions of casein micelles. Milk pH in the mammary glands is neutral, but in neonatal stomachs, the pH environment of the milk first drops rapidly and then increases slowly as milk exits the stomach and enters the intestines [17–19]. pH plays a significant role in the aggregation of casein micelles by dissolving the CaP and altering protein interactions [3,20]. In addition, this decrease in pH better suits chymosin which has a higher optimum pH than pepsin.

The kinase Fam20C (EC 2.7.11.1) phosphorylates caseins in the Golgi apparatus of mammary epithelial cells (MEC). It phosphorylates serines followed by a negative charge residue according to the `Ser-x-Glu/pSer` motif [21–23]. Some kappa-casein chains are also O-glycosylated [4]. Unlike kinases, glycosyltransferases do not recognize motifs that can be reduced to a simple regular expression [24]. Kappa-caseins chains from an identical milk sample can have different numbers of phosphorylations and glycosylations [4], thus altering the potential charge and aggregation propensity of micelles.

Two models of casein micelle formation have been proposed, one suggesting that polar interactions may be critical [25], and the other placing a greater emphasis on hydrophobic interactions [26]. We set out to systematically study the evolution of kappa-caseins, which play such a pivotal role in determining micellar structure and function. Our analysis of kappa-casein primary sequence establishes a model of kappa-casein evolution in which $\pi$-interactions are hypothesized to play an important role, and in which the casein micelle, despite having originated at the base of mammalian clade, is still undergoing evolutionary change impacting on its function.

# 2. Material and methods

## 2.1. Alignment of kappa-casein protein sequences

Complete kappa-casein sequences from mammals were downloaded from the NCBI's protein database (https://www.ncbi.nlm.nih.gov/protein). We discarded protein sequences with missing exons. When

multiple isoforms were available in a species, only a representative was selected (electronic supplementary material, table S1). The complete sequence from the American manatee (*Trichechus manatus*) was included, identified via a BLAST search using the second exon of the elephant (*Loxodonta africana*) kappa-casein sequence (see electronic supplementary material; [27]). We gathered a total of 99 sequences of kappa-casein sequences from placental, marsupial and egg-laying mammals. To remove the signal peptide in each sequence, the protein sequences were aligned using MAFFT [28] with the known position of the signal peptide in cattle. The alignment of the mature sequences was manually modified to correct issues with indels and protein tandem repeats (see electronic supplementary material), and to define tandem repeat boundaries manually for insertions of three or more residues. To contrast the two parts of kappa-casein, para-kappa-casein (PKC) and glyco-macro-peptide (GMP), we used the known position of the cleavage site in cattle to assign these regions in other species.

## 2.2. Mammal phylogeny and maximum-likelihood protein tree

The supertree of 5020 extant mammals from Fritz *et al.* [29] provided the tree topology along with divergence times in millions of years. The use of this species topology was justified by the observation that kappa-casein appears to be encoded by a single gene without validated evidence of duplications in any species. Species names were modified to their binomial form and to match those of the supertree. The supertree was pruned to remove species without an available kappa-casein sequence using the R package 'ape' [30]. Species with a kappa-casein that were absent from the supertree were placed at the position of a closely related species (electronic supplementary material, table S1). We used the topology of this tree to build a maximum-likelihood protein tree optimizing the branch length with optim.pml from the 'Phangorn' R package [31]. We determined, using the modelTest function implemented in 'Phangorn', that the Jones–Taylor–Thornton (JTT) evolutionary model with a gamma distribution was best [31–33]. To compare species and protein divergence, we plotted the sequence pairwise distance and the divergence time, for every species pair.

## 2.3. Conservation

To score the conservation of a residue at a given position, we calculated the fraction of sequences for which each amino acid was found at each position, weighted according to branch lengths. For this weighting, we used the Gerstein–Sonnhammer–Chothia (GSC) algorithm, implemented in the aphid R package, on the pruned species supertree [34,35].

## 2.4. Prediction of physico-chemical properties and disorder

For the mature sequence and both parts of each kappa-casein, we computed the grand average of hydropathy (GRAVY) using the Kyte–Doolittle amino acid scale of hydrophobicity [36]. Similarly, we computed the net charge from pH 1.5 to 7.5 with the 'Peptides' R package using the pK values from the Bjellqvist scale [37,38]. The prediction method IUPred2 with the 'long' parameter was used to predict the likely disorder for each position of each sequence [39].

## 2.5. Prediction of phosphorylated and O-glycosylated residues

To analyse the conservation of phosphorylations, we predicted for each sequence the positions of serines matching the canonical Fam20C phosphorylation motif [21]. We transformed this motif into the regular expression: `(?<=E.(S.){0,n})S|S(?=(.S){0,n}.E)`. This regular expression matches `Ser-x-Glu` and clusters of phosphorylated serines `Ser-x-(Ser-x)n-Glu`, with the serine in the middle repeated `n` times (we arbitrarily chose n = 10). This regular expression matches serines in both directions. More complex methods need to be used to predict O-glycosylated serines and threonines due to the lack of any simple recognised motif. We tested different published machine learning software: NetOGlyc4.0 [40]; GlycoMine [41]; GlycoPred [42]; and O-GlcNAcPRED-II [43]. We used UniCarbKB incorporated in GlycosuiteDB to find known glycosylation sites for human and cattle kappa-casein [44,45]. Comparing the predicted sites with the known sites, we found that GlycoMine performed better than the other software, with the highest precision, accuracy and Matthews correlation coefficient (MCC), and with the second-best sensitivity (electronic supplementary material, table S3; [46]). Accordingly, we used it in the full analysis.

## 2.6. Predictions of plasmin and pepsin cleavage sites

For each sequence, we used the 'cleaver' R package to predict cleavage sites positions of the endopeptidases plasmin and pepsin. This package uses regular expressions to find possible cleavage sites. For pepsin, it uses the cleavage rule described in ExPASy PeptideCutter, and for plasmin, it cleaves after Arginine and Lysine [47,48].

## 2.7. Statistical analyses

Data analysis was performed using the R environment v. 3.4.4 [49]. The study made extensive use of the Tidyverse R package suite [50] and of the 'ProjectTemplate' [51], 'ggtree' [52], 'Biostring' [53] and 'ape' [30] packages.

# 3. Results

Understanding the evolution of kappa-casein is important to better understand the structure and function of casein micelles. Lemay *et al.* [54] showed that caseins evolve more rapidly than other milk proteins. Different research groups previously compared casein sequences from a few different species [2,55–58]. Here, we systematically investigated the relationship between structure and function, paying particular attention to the contrasting influences upon the PKC and GMP regions.

## 3.1. Rapid and slow phases of kappa-casein evolution

Kappa-casein's origins are ancient in mammalian evolution. Its gene is present in all mammalian lineages, along with beta- and/or alpha-caseins. It might therefore be expected that the functional constraints were largely determined at an early stage, before the separation between therians and prototherians (166 Ma; Fritz *et al.* [29]).

We aligned the mature protein sequence of kappa-casein (electronic supplementary material, figure S1) and compared the rate of sequence change over different epochs of mammalian evolution, by contrasting the pairwise amino acid differences between all sequence pairs with the inferred timescale of evolution. Unsurprisingly, more distant clades show the highest degree of divergence. However, unexpectedly, the rate of sequence change seems to have been much greater in deeper branches (60–170 Ma) and then to have slowed in the last 60 Myr (figure 1b). To ensure that this was not simply a technical artefact of an over-correction for multiple substitutions employed during estimation of the number of replacements, we repeated the analysis without correction for multiple testing, and observed a similar slow-down (electronic supplementary material, figure S2). While there are a number of potential explanations for this observation, it is consistent with the hypothesis that kappa-casein structure and function had not yet stabilized during the more ancient epoch, such that it was undergoing considerable changes in multiple lineages. We thus suggest a potential two-epoch model for kappa-casein evolution of adaptation (170–60 Ma) followed by relative stabilization (0–60 Ma).

### 3.1.1. Protein tandem repeats indicate stronger length constraints on the PKC.

While the phylogeny-weighted average length of the mature kappa-casein is 164.2 residues, there is considerable variability in total length, ranging from 145 residues in short-beaked echidna (*Tachyglossus aculeatus*) to 237 residues in mouse-eared bats (*Myotis lucifugus* and *Myotis brandtii*). Some of this sequence difference reflects a gradual increase in average length from the egg-laying mammals (platypus and echidna, average length = 146.0) to marsupials (average length = 158.0) to placental mammals (average length = 165.7). Short indels (up to eight residues) appear to have occurred after the split between Monotremata and Theria and after the split between Marsupialia and Placentalia (electronic supplementary material, figure S3). The most striking changes are the result of large insertions within the placental mammals, with these insertions predominantly confined to the GMP (figure 2). A second notable feature of these large insertions is that they reflect tandem duplications of two to three copies of particular regions (figure 2). In addition to the guinea pig's (*Cavia porcellus*) tandem repeat (TR) noted by Hall [59], we found TRs in 10 different species, using the alignment as reference (electronic supplementary material, table S2). The fact that large insertions are limited to the GMP (figure 2) indicates that the length of PKC is relatively constrained, probably

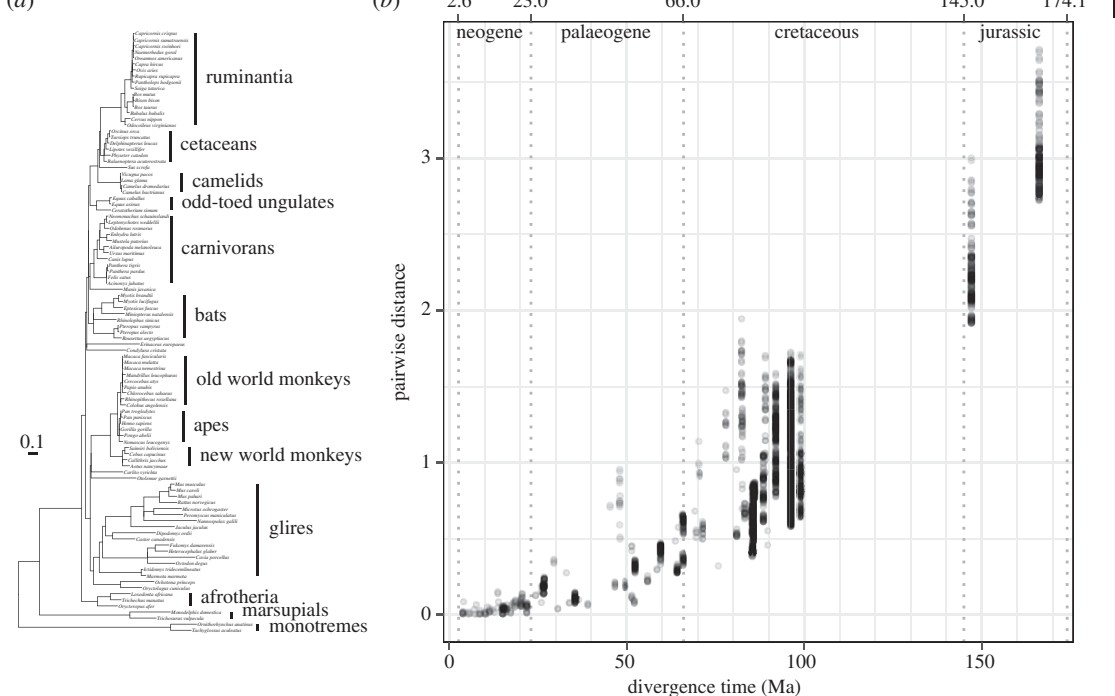

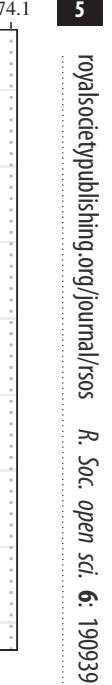

**Figure 1.** Clade-specific and time-period specific variation in the rate of evolution of kappa-casein. (*a*) Maximum-likelihood protein tree of kappa-casein using the topology of the species [29]. The two monotremes, platypus (*Ornithorhynchus anatinus*) and echidna (*Tachyglossus aculeatus*), are used to root the tree (See Material and methods). The scale bar represents amino acid replacements per site. (*b*) Scatter plot of the pairwise distance between mature kappa-casein sequences (maximum-likelihood JTT + G model) and the divergence time.

by its role in stabilizing the casein micelle internal structure. This is consistent with the model in which the GMP is exposed on the exterior of the micelle, such that gross changes in length are less likely to disrupt micelle formation or stability. However, there have been no large deletions in the GMP, suggesting that there may be a minimal length requirement, or that the different subregions each play key roles which prevents their deletion. In one particular lineage, the one leading to the northern greater galago (*Otolemur garnetti*), there has been not just one, but two separate tandem repeat duplications, consistent with a positive selection pressure favouring a longer GMP sequence, resulting in its amplification in two separate regions. Large insertions are not all very recent evolutionary phenomena, with subsequent substitutions and indel events having occurred after TR duplication in the beaver (*Castor canadensis*), the guinea pig (*Cavia porcellus*), the northern greater galago (*Otolemur garnettii*) and the mouse-eared bats (*Myotis*). In the little brown bat (*Myotis lucifugus*) and in the Brandt's bat (*Myotis brandtii*), which are closely related mouse-eared bats, a sequence of 37 residues is repeated twice, spanning the cleavage site for chymosin, which may impact on the rate and efficiency of cleavage. It appears that these TRs are relatively evolutionarily stable, and therefore unlikely to confer any disadvantage.

### 3.1.2. Conservation of compositional biases highlights which amino acids are key to GMP and PKC function

Kappa-casein is known to have regions of particular amino acid compositional enrichment [58]. These compositional preferences have been used in part to favour particular models of kappa-casein structure and function [58,60]. We investigated evolutionary variation in the composition of different amino acids in the PKC versus the GMP, and how they might relate to the function of kappa-casein. The range of frequencies varied considerably for certain amino acids, with particular contrasts between the PKC and the GMP (figure 3).

Serines (S) can be both phosphorylated and O-glycosylated, but are observed at lower levels than threonines (T) in the GMP. Threonine can be O-glycosylated and has the highest abundance in many species (electronic supplementary material, figure S4), suggesting a functional preference for O-glycosylation over phosphorylation in this region. Conversely, serine is greatly favoured over threonine in the PKC. The wide variation (10–30%) in threonine composition suggests that there may

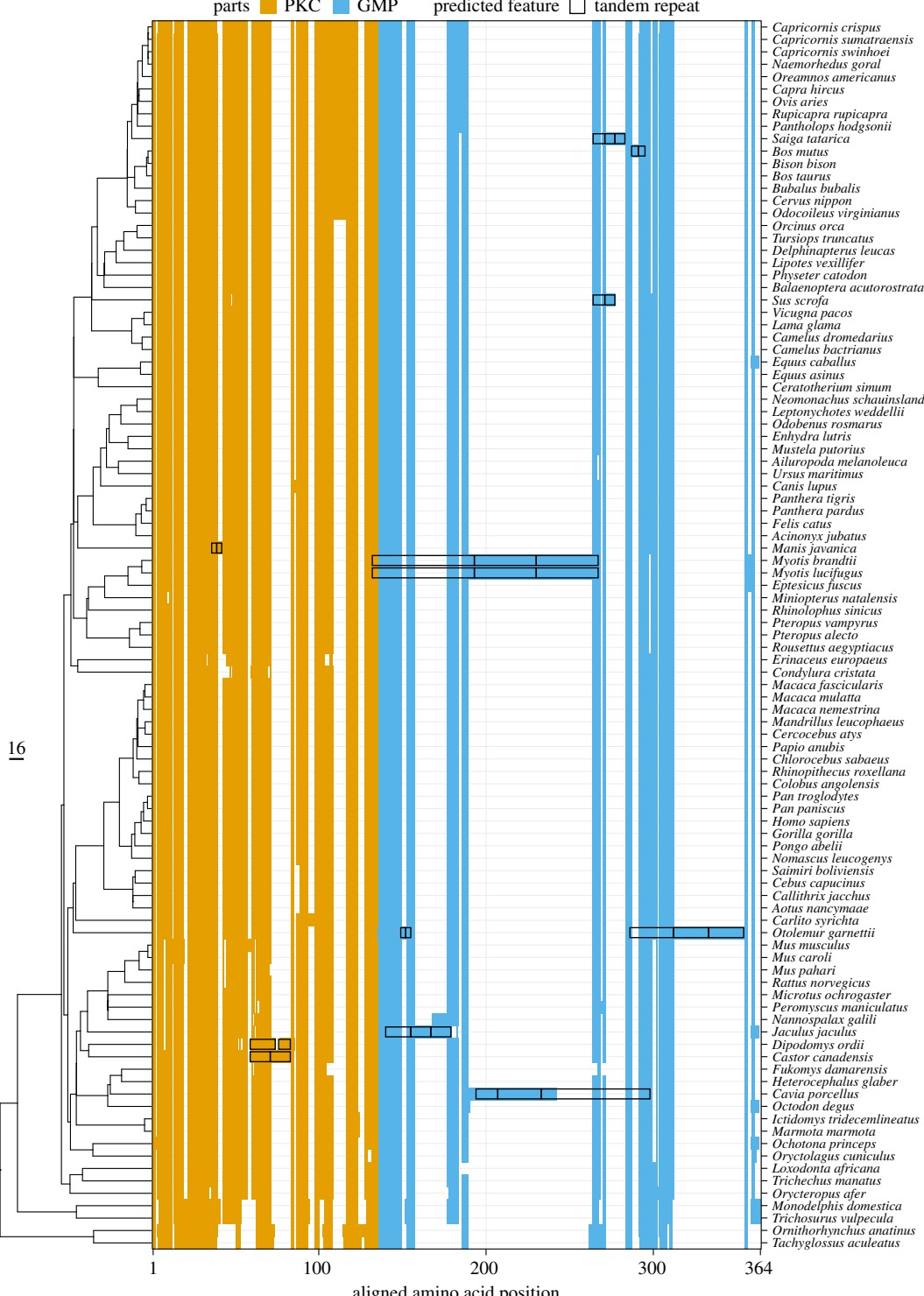

**Figure 2.** Length variation via indels and protein tandem repeats in PKC and GMP. On the side, a mammalian species tree is displayed [29]. Aligned residues kappa-casein sequences are showed in orange for the para-kappa-casein (PKC) and in blue for the glyco-macro-peptide (GMP). Gaps in the alignment are showed in white. The black boxes represent manually annotated tandem repeats.

be a consequent high variability among species in the degree of glycosylation (see below). For two other polar amino acids, asparagine (N) and glutamine (Q), the distribution of the range of compositions suggests that they are not strongly confined to either the PKC or the GMP. Aromatic amino acids are rare or absent in the GMP, whereas tyrosines (Y) and phenylalanines (F) are present one or more times in the PKC, up to 10% in certain species (electronic supplementary material, figure S4).

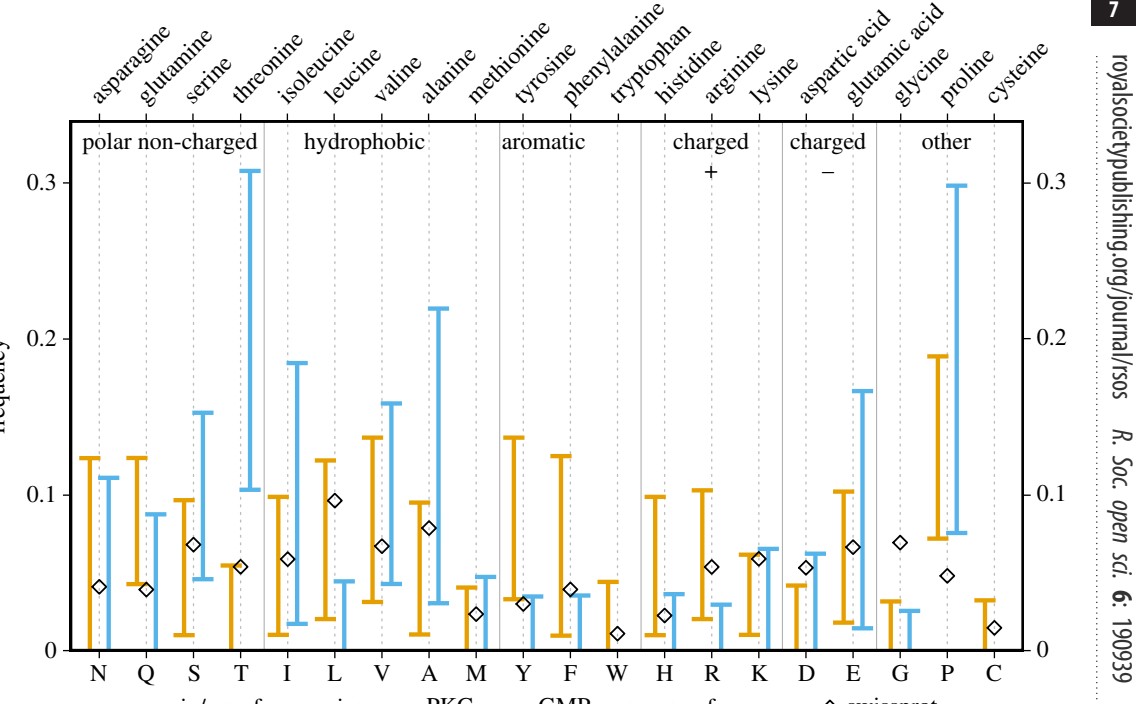

**Figure 3.** The ranges of possible amino acid frequencies in PKC and GMP. Each range show the minimum and maximum frequency of each amino acid in para-kappa-casein (PKC; in orange) and glyco-macro-peptide (GMP; in blue) sequences (the full data are presented in electronic supplementary material, figure S4). For each amino acid, the black lozenge represents its average frequency in SwissProt [61].

The distribution of positively charged residues differs strikingly between PKC and GMP. Arginines (R) and histidines (H) are almost exclusively found in the PKC, while lysines (K) are found in both parts of the protein. The functional basis for the avoidance of arginine and histidine in the GMP and their preference in the PKC is unknown. It seems unlikely to be determined by preferences of relevant proteases important in milk digestion, since both plasmin and trypsin cleave at arginine and lysine, but not after histidine. We hypothesize that arginines may be preferred over lysines in the PKC to form cation-$\pi$ interactions [62]. In contrast to the positively charged amino acids, the negatively charged amino acids do not show marked differences in their PKC/GMP preferences (figure 3).

Leucine (L) is rare or absent in the GMP, where isoleucine (I) can be found in significant numbers. In the PKC, both residues are found in similar quantities. We note that leucine is more abundant than isoleucine in protein sequences [61]. This suggests that isoleucine may perform a particularly useful role in the GMP, perhaps involved in some aspect of protein–protein interaction (see below for a discussion of the role of protease interactions). An alternative metabolic hypothesis is that the GMP is digested earlier, along with soluble whey proteins, while the PKC is sequestered temporarily in casein aggregates that are more gradually accessible to digestive proteases. This would release isoleucine earlier, and isoleucines may have a signalling advantage over leucine in triggering responses of metabolism controlling cells [63]. Cysteine (C) is found exclusively in the PKC, but in different numbers in different species (see below for a discussion about possible disulphide bond gains and losses).

In summary, a number of amino acids show clear preferences for GMP or PKC. We can explain some of these biologically, such as the threonine preference in GMP relating to glycosylation. However, others highlight aspects of kappa-casein which merit further analysis to investigate their functional or structural roles, such as the preference for isoleucine in the GMP and the avoidance of histidine and arginine in the GMP.

## 3.2. Conservation of residues, charge, disorder and hydrophobicity

Certain amino acids enriched in casein sequences may play important functional roles at particular residue positions. We examined the degree of evolutionary constraint for each amino acid at each position, weighting the analysis according to the evolutionary tree branch lengths. A number of interesting

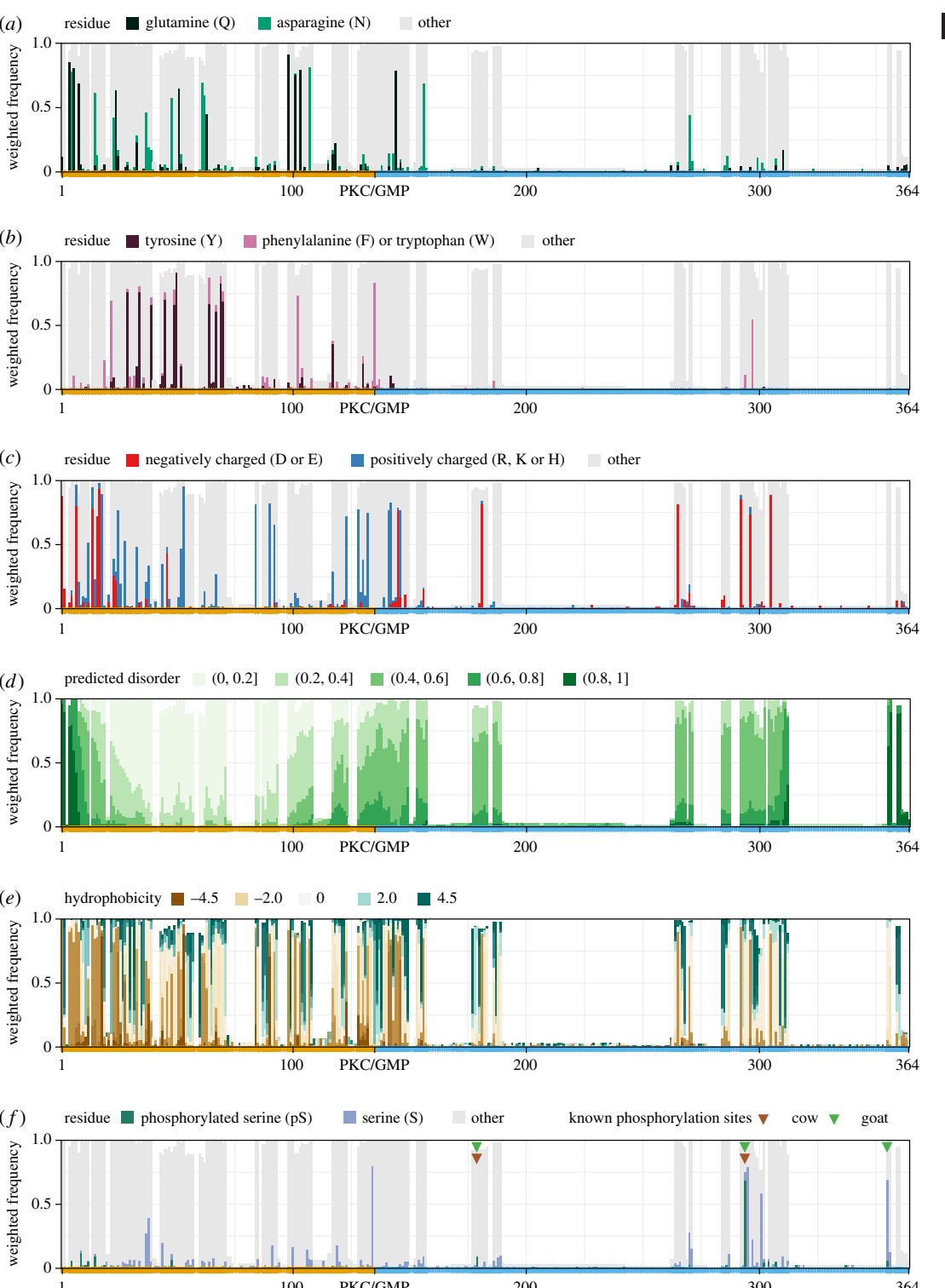

**Figure 4.** Tree-weighted evolutionary conservation for amino acid and motif properties. (*a*) Asparagine/glutamine; (*b*) aromatic; (*c*) charged; (*d*) disorder predicted with IUPred2A [39] grouped in groups of 0.2; (*e*) hydrophobicity values using the Kyte–Doolittle scale [36]; (*f*) FAM20C predicted phosphorylation. Gaps in the alignment are not represented and appear as white in all plots. Below each plot, the orange line represents PKC, and the blue line represents GMP. Coloured triangles: experimentally identified phosphorylated residues in goat (*Capra hircus*) and cow (*Bos taurus*) [64,65].

findings emerged. While glutamine (Q) has been postulated to play a key role in the PQ-rich region in the PKC [3], we found that glutamine was not particularly conserved at particular positions, when compared with the similar amino acid, asparagine (figure 4*a*), even though some proteins completely lack asparagine

while none completely lack glutamine (figure 3). PKC tyrosine residues showed a higher degree of conservation, indicating a potentially more critical functional role in determining the structure of casein micelles (figure 4b). Charged residues also showed a high level of conservation of charge type, suggesting that the locations of charged residues contribute to the actual structural conformation of caseins, in addition to their contribution to the net charges of the PKC and GMP regions (figure 4c).

In the GMP, conserved charged residues are mainly negative, with a few positively charged conserved residues near the cleavage site at the start of the GMP. While the structural constraints in the micelle core may explain charge positional conservation in the PKC, it is unclear why the positional conservation of the negatively charged residues in the GMP is so strong. This is suggestive of their potential role in interacting with the micelle exterior or with other components. The likely degree of disorder of residues (predicted by IUPred2A; [39]) prior to cleavage indicates that disorder is conserved at the two termini of the mature protein (figure 4d). The central region of the PKC shows a strong conservation of the most ordered category, suggesting that this region may adopt a stable tertiary structure within the micelle. While hydrophobicity might be considered to be an important property in facilitating such structure, there are no clearly conserved extensive patches of hydrophobicity, with the more extensive conserved patches of hydrophilicity (low-scoring regions in brown, figure 4e) dominating the conservation landscape of the protein. This suggests that the conservation of hydrophilic patches is more important for PKC function than the conservation of hydrophobic regions. Thus, whatever structures may form within the micelle may be relatively accessible to the solution, and are less likely to form a hydrophobic core, typical of ordered proteins. This is consistent with the characterization of kappa-casein as a disordered protein when in solution [66].

## 3.3. High species variability of GMP predicted net charge at neutral pH

We computed the net charge per residue (NCPR) at different pH for the mature sequences and for the PKC and GMP separately using the Bjellqvist's pK scale as coded within the 'Peptides' R package [37,38]. This is based on the primary amino acid sequences and ignores additional charge features imposed by post-translational modification. The NCPR values shown in figure 5 will be shifted to more negative values after both phosphorylations and glycosylations, which in milk contain sialic acid [67]. With the acidification of milk in the stomach, first glutamate ($pKa = 4.05$) and aspartate ($pKa = 4.25$) are neutralized, followed by the phosphate groups ($pKas$: 2.2, 5.8 and 12.4); and sialic acid ($pKa = 2.2$) present in O-glycosylation [37,68,69]. While it is possible to calculate the charge differences resulting from such modifications, it is difficult to predict the overall efficiency of modification, which may itself vary between species. We therefore focus here on the relative differences in charge states predicted for the primary sequences without modification.

On average, at pH 7, mature kappa-casein in the absence of post-translational modifications are neither positively nor negatively charged (average NCPR = 0.00, s.d. = 0.02), although a few outliers have either positive or negative charge (NCPR ranging from −0.07 in guinea pig *Cavia porcellus* to 0.04 in black-capped squirrel monkey *Saimiri boliviensis*, figure 5a). This is consistent with the findings of Khaldi & Shields [70]. As expected, a decrease in pH increases the NCPR of the mature primary sequence (figure 5a). At pH 3, kappa-caseins are all positively charged (average NCPR = 0.10, s.d. = 0.01).

At pH 7, the PKC sequence is positively charged (average NCPR = 0.03) while the GMP sequence is negatively charged (−0.06). The variation in net charge of the GMP at pH 7 is greater than at low pH, reflecting in part changes due to large insertions within certain species. Thus, at pH 7, some primary GMP sequences are effectively uncharged at pH 7 (e.g. black-capped squirrel monkey *Saimiri boliviensis*; NCPR = 0.01) and others have a very strong negative charge (e.g. guinea pig *Cavia porcellus*; NCPR = −0.15). These differences become less pronounced at a low pH. Mice and the prairie vole (*Mus* and *Microtus ochrogaster*) are outliers at pH 3 with an almost neutral primary GMP sequence while it is positively charged in other species (figure 5b). This may relate to the absence of positive charge at the N-terminus of the GMP of these species.

## 3.4. Lack of conservation and potential absences of phosphorylation

We focused on evolutionary analysis of canonical phosphoserine predictions (Ser-x-(Glu/pSer), calculated iteratively) to prevent a high number of false-positive predictions. While eight sites were seen in the guinea pig's GMP, we do not predict phosphorylation sites anywhere in kappa-casein in Old World monkeys, apes, seals and ground squirrel. No phosphorylation sites were predicted in the GMP of egg-laying mammals, platypus and echidna, but multiple clustered phosphorylations are predicted in

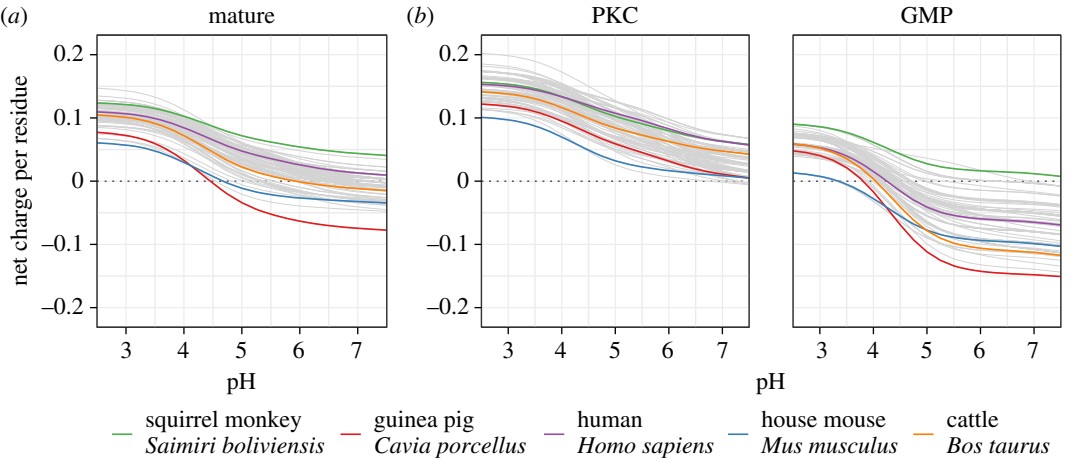

**Figure 5.** Evolutionary variation in the net charge per residue (NCPR) between pH 2.5 and 7.5. (a) mature kappa-casein primary sequence, (b) PKC and GMP primary sequences. Each species is represented by a different line. We use the pK values from Bjellqvist's scale [37].

the PKC (electronic supplementary material, figure S5). Overall, this lack of conservation indicates that phosphorylation by Fam20C is unlikely to contribute to conserved structural features of the protein, and instead may contribute beneficially to the overall charge properties of the protein. It is likely that, in species with lower levels of Fam20C phosphorylation, the pH properties are compensated for by other features in the primary sequence or in the glycosylation patterns of the GMP. It is also possible that phosphorylation of kappa-caseins may have other functions such as antimicrobial activity [71]. Conservation of the prediction requires conservation of both the serine and an adjacent glutamic acid. We noted that serines, in general, are poorly conserved in both the PKC and the GMP (figure 4f). In addition, most of these predicted phosphorylation sites are present in only a few species or clades (average weighted fraction of species = 0.06). One of the three experimentally validated phosphorylation sites is, however, strongly conserved (S149 in mature bovine kappa-casein), being predicted in 69 of the 72 species. Interestingly, this strongly conserved predicted serine is flanked by two glutamic acids, one on each side. While this site has been found to be phosphorylated [13,65,72,73], there is evidence that it may also be O-glycosylated in cattle [73]. This predicted phosphorylation site is duplicated within tandem repeats in northern greater galago (*Otolemur garnettii*), guinea pig (*Cavia porcellus*) and yak (*Bos mutus*). Despite the apparent importance of this serine, it is absent in kappa-casein of some individual species and of the following taxa: monotremes, marsupials, old world monkeys, apes, ground squirrels and seals. Thus, overall predicted phosphorylations are generally poorly conserved and the most strikingly conserved predicted phosphorylation is absent from a number of species, including some with no predictions whatsoever throughout the protein (figure 6). This indicates that phosphorylation may not be critical to kappa-casein function, in either a position-specific or position-independent fashion.

## 3.5. Avoidance of para-kappa-casein glycosylation

We analysed the kappa-casein sequences with GlycoMine, choosing a threshold of 0.4 to optimize sensitivity, thus predicting most of the known glycosylation sites in human and cattle. The predicted sites are almost entirely confined to the GMP, indicating that glycosylation sites are actively avoided in  the PKC, where glycosylation is likely to disrupt micelle formation (figure 4g; electronic supplementary material, figure S5). The number of predicted glycosylation sites in the GMP ranged from 1 to 20 (figure 6), suggesting that at least one glycosylation is likely to be necessary for function, but that the degree of glycosylation is highly variable between species. It is not surprising to find differences between human and bovine predictions, as important differences between human and bovine glycomes were experimentally described [74–76].

## 3.6. Independent predicted disulphide bond acquisition during evolution

In mature bovine kappa-casein, there are two cysteines, which can form covalently bound oligomers and intrachain bonds [77,78]. Since not all mammalian species have these cysteines, different disulphide bonding patterns are expected with possible consequences for casein micelle stability. For many

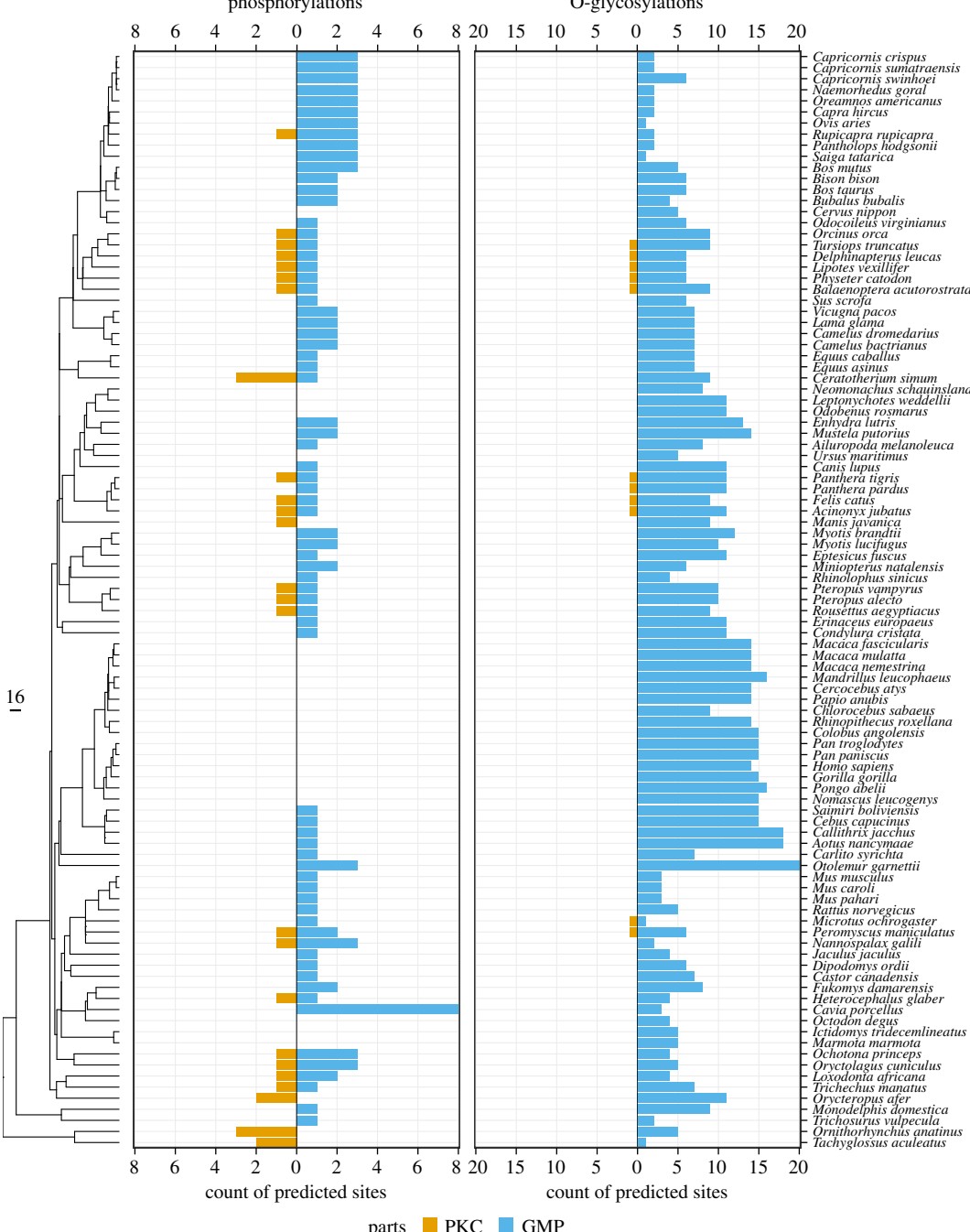

**Figure 6.** Predicted phosphorylations and glycosylations in PKC and GMP. Sequences, on the *y*-axis, are ordered according to their phylogenetic relationships based on the supertree [29]. The number of predicted post-translational modifications is displayed for each sequence for para-kappa-casein (in yellow, from right to left) and glycomacropeptide (in blue, from left to right). Phosphorylation predictions were performed using a regular expression of the Fam20C canonical motif. Glycosylation predictions were performed with GlycoMine (cut-off = 0.4) [41].

species with only one cysteine, such as human, only dimers can form, but for sequences with two or three cysteines more complex linkages are possible, depending on the number of cysteines and their positions (electronic supplementary material, figure S6). Kappa-casein sequences with two non-sequentially adjacent cysteines can form intrachain bonds or alternatively inter-chain oligomers. Kappa-caseins with two consecutive cysteines can only form oligomers. The addition of a more distant cysteine in the PKC allows the additional formation of intrachain bonds.

All therian mammals (placental and marsupial mammals) have mature kappa-casein sequences containing at least one cysteine aligned with either position 10 or 11 in cow (figure 7). Disulphide

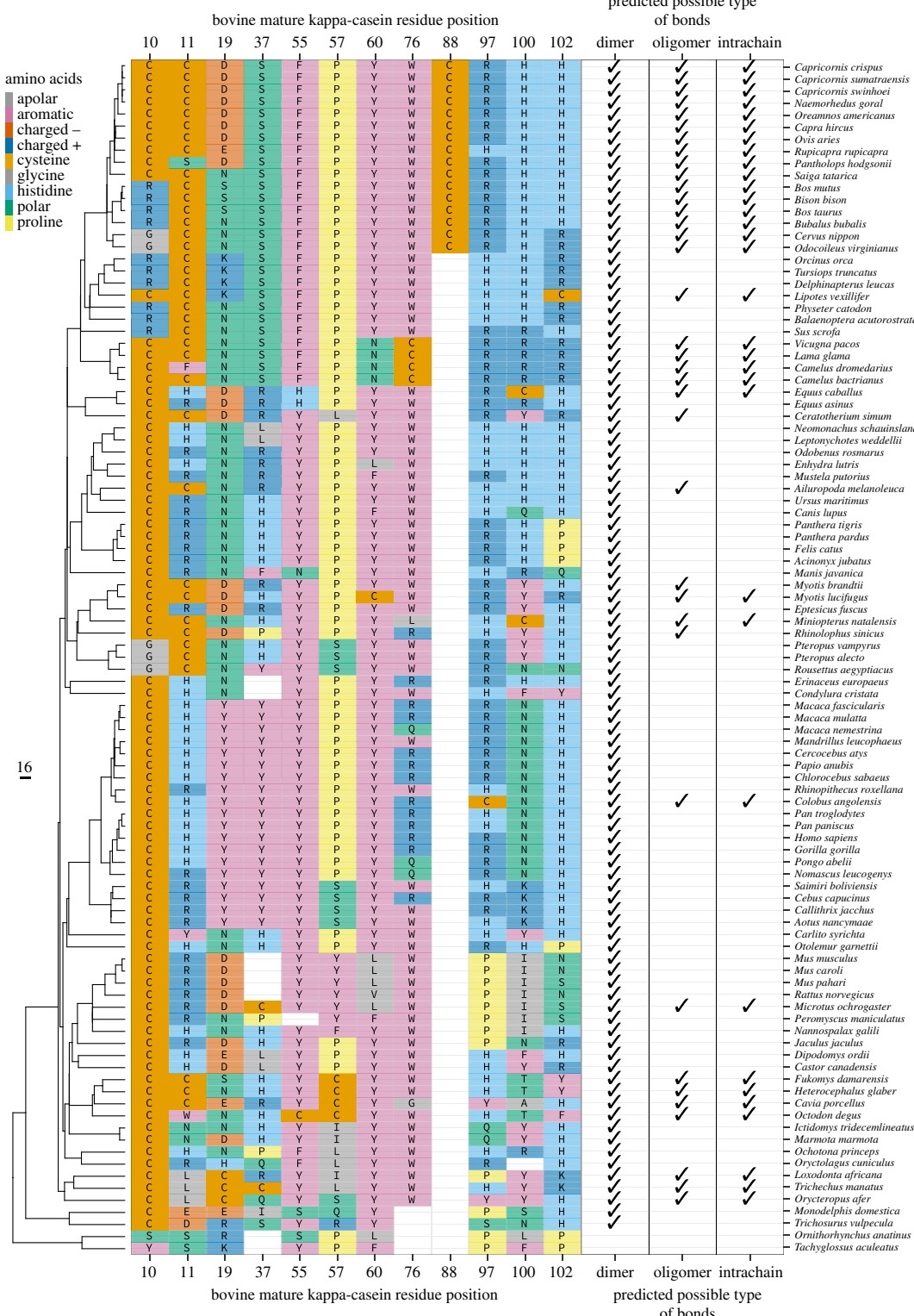

**Figure 7.** Evolution of cysteine positions and potential disulphide bonding patterns. Left: the phylogenetic tree of mammals (distance in Ma), trimmed from Fritz *et al.* [29]. Main panel: residues present at alignment positions where cysteine is observed in any species. These are coloured according to their properties and numbered according to bovine mature kappa-casein. Right: possible scenarios of kappa-casein covalent bonding in each species. Intrachain: two cysteines separated by at least one residue. Dimer: at least one cysteine. Oligomer: at least two cysteines.

bonds may possibly improve thermal stability, since placentals and marsupials have higher body temperatures than monotremes [79]. The proposed ancestral origin of kappa-casein, the follicular dendritic cell secreted peptide (FDCSP), as well as the sequences of egg-laying mammals' kappa-

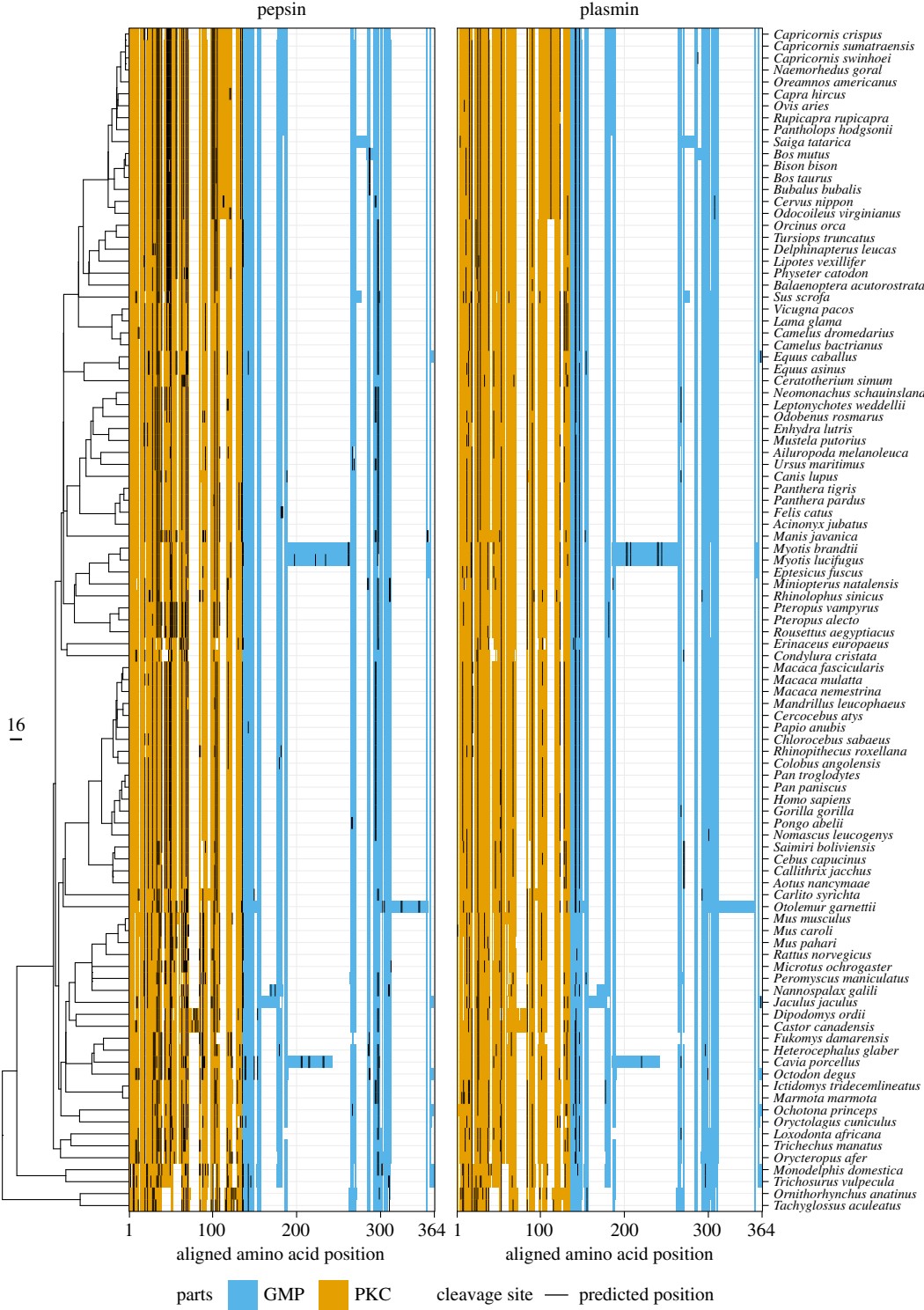

**Figure 8.** Distribution of pepsin and plasmin cleavage sites in kappa-casein sequences. The vertical line represents the predicted cleavage site. Aligned residues kappa-casein sequences are showed in orange for the para-kappa-casein (PKC) and in blue for the glyco-macro-peptide (GMP). Gaps in the alignment are showed in white. On the side, the phylogenetic tree of mammals (distance in Ma), trimmed from Fritz *et al.* [29]. Regular expressions from ExPASy PeptideCutter were used to predict cleavage sites with the R package Cleaver [47,48].

casein do not contain cysteines [2,80]. Thus, they are not able to form any disulphide bonds. This suggests that after the split from egg-laying mammals, 166 Ma [29], therian mammals' kappa-casein gained the ability to form dimers. This ability was never lost despite all positions being subject to some mutations, suggesting that dimer formation confers an evolutionary advantage. Additional to

10/11 position cysteines, other cysteines appear and disappear at various positions within the PKC, which may confer stability through disulphide bond formation. Cysteines are completely absent from the GMP in all species, suggesting that cross-linking would be likely to disrupt micelle formation, either through limiting the freedom of the GMP to interact with the aqueous environment, or through the formation of disulphide bonds between micelles, promoting premature deleterious aggregation.

A number of lineages have independently evolved to have more than one cysteine that are not immediately adjacent to each other. None of those lineages have ever evolved back to having only one cysteine, despite the fairly continuous turnover of cysteines during evolution. These extra cysteines may occur at various positions along the PKC, in addition to those at the N-terminus. This indicates that some advantage is conferred by the greater complexity of disulphide bond formation involving two cysteines. Whether the advantage is in shifting from dimerization to oligomerization, or in the formation of intrachain disulphide bonds is unclear, and both are known to exist in bovine kappa-casein [78]. While a number of groupings evolved to have two cysteines at the 10/11 positions, these have only been subsequently lost during evolution when there is another cysteine elsewhere in the protein. This suggests that the ability to oligomerize may be one important advantageous feature of multiple cysteines.

These patterns of accumulating complexity over evolutionary time indicate that differences in the number and position of cysteines are likely to influence the structure and stability of the casein micelle. A likely evolutionary transition from no linkages, to dimerization, to oligomerization, to intrachain bonding capability are suggested by these findings (figure 7, right panel). Each of these linkages is likely to have an impact on the integrity of casein micelles, as suggested previously [66].

## 3.7. Glyco-macro-peptide sequences lack protease cleavage sites

We investigated the cleavage sites for pepsin and plasmin (see Material and methods). Plasmin is a major peptidase found in milk both in its proenzyme and active forms [14]. It preferentially cleaves after positively charged residues. Trypsin, a major intestinal peptidase, has a similar but slightly more stringent cleavage specificity than plasmin. This cleavage preference differs from the gastric peptidases chymosin and pepsin, which preferentially cleave before or after the aromatic residues phenylalanine, tyrosine and tryptophan, and leucine with chymosin having a slightly more specific substrate specificity [10,47,81]. PKC has a much greater density of cleavage sites per residue than GMP (paired one-tailed $t$-test; $p$-value $<10^{-15}$), where cleavage sites are either rare or absent (figure 8). This may explain in part the preference of isoleucine in the GMP over leucine, as this avoids pepsin cleavage. Thus, if all the isoleucine in bovine kappa-casein were replaced with leucine, the number of pepsin cleavages in the GMP would rise from 2 to 9 sites. A possible explanation for the avoidance of cleavage sites in the GMP is to prevent premature aggregation of milk, in particular by plasmin, which is itself present in the milk, as well as by other milk proteases, such as cathepsin D [14], which share similarities with the pepsin hydrophobic cleavage preferences, and a preference for cleavage after leucine [82].

# 4. Conclusion

The role of kappa-casein in casein micelle formation and aggregation in different species is not completely understood, and the features determining its casein interactions are not well defined. To address this, we performed an evolutionary analysis of the primary sequence of this fast evolving protein in 99 mammalian species.

While Holt *et al.* [3] emphasized the role of polar interactions, the evolutionary positional conservation constraints that we noted in the PKC appear to emphasize the importance of positively charged and tyrosine residues. Residue enrichments were also consistent with likely PKC phosphorylation and GMP glycosylation modifications, which in turn may dominate many aspects of function. In addition, we noted the avoidance of protease cleavage sites in the GMP. A key component that may impact on the stabilization of casein micelles are disulphide bonds. We noted the PKC's increased capacity for potential disulphide multimerization appearing in independent lineages over evolutionary time, suggesting that casein interactions may be still capable of further functional optimization.

The caseins in a casein micelle are likely to be substantially disordered, in the sense that they do not adopt a fixed structure, and move freely while maintaining an overall shape and coherence of the micelle. The intrinsic disorder of casein may have allowed a rapid evolution of these proteins and contributed to the evolution and variations of milk components and functions in different mammalian species. This lack of structural constraint may have contributed to the rapid evolution of kappa-casein, which was

particularly notable prior to 60 Ma, and permitted divergence in its composition among different mammalian species. Sequence disorder can be associated with polar tracts, with polyampholyte tracts (lots of both positive and negative charge) or with polyelectrolyte tracts [83]. While short polar and polyelectrolyte tracts are seen in kappa-casein, evolutionary conservation suggests that charge is more critical than polarity. Both the modifications of kappa-casein, of phosphorylations in the PKC and glycosylations in the GMP, tend to increase negative charge. Thus, it may well be that polyelectrolyte effects dominate kappa-casein's structure and interactions in the micelle.

An alternative model to Holt *et al.* [3] is that of Horne [60]. This model places an emphasis on hydrophobic residues. While in kappa-casein we noted no strong evolutionary constraint on hydrophobicity in itself, we noted that tyrosines showed particularly strong positional conservation in the PKC. Their role is likely to be more than a simple hydrophobic effect. Aromatic rings can play key roles in CH–$\pi$, cation–$\pi$ and $\pi$–$\pi$ interactions, and the PKC shows conservation of both aromatic and positively charged residues. Such $\pi$ interactions are analogous to the protein interactions involved in phase transition and membraneless organelle formations [84–86].

Thus, our in-depth analyses of the evolutionary dynamics of kappa-casein suggest multiple constraints on protein interaction, additional to those previously proposed. It is likely that multiple physico-chemical forces combine to achieve the optimal structural behaviour for this critical contributor to mammalian nutrition and survival.

Data accessibility. The code and data are available within Zenodo https://doi.org/10.5281/zenodo.2587122 [87]. The following supporting files are available as electronic supplementary material: figures S1–S6 and tables S1–S3 (pdf), American manatee (*Trichechus manatus*), Kappa-casein protein sequence (fasta Kappa-casein protein alignment).
Authors' contributions. J.M. and D.C.S. designed the research, interpreted the data and wrote the paper; J.M. conducted the analysis. All authors have read and approved this manuscript.
Competing interests. We have no competing interests.
Funding. Funded by Enterprise Ireland grants to Food for Health Ireland (TC2013001, TC20180025).
Acknowledgements. We thank Dr Norman Davey and Chiara Cotroneo for useful discussion.

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
