## [Reviewer comments · Royal Society Open Science]

Review History

RSOS-190939.R0 (Original submission)

Review form: Reviewer 1

Is the manuscript scientifically sound in its present form?

Yes

Are the interpretations and conclusions justified by the results?

Yes

Is the language acceptable?

Yes

Do you have any ethical concerns with this paper?

No

Have you any concerns about statistical analyses in this paper?

No

Recommendation?

Accept as is

Comments to the Author(s)

Manuscript Number: RSOS-190939

Title: Implications of kappa-casein evolutionary diversity for the self-assembly and aggregation of casein micelles

Article Type: Research Article

Dear Editor and Authors

I checked the manuscript: RSOS-190939, entitled "Implications of kappa-casein evolutionary diversity for the self-assembly and aggregation of casein micelles" Manguy and Shields. In this work, the evolution of kappa-caseins has been systematically study, since it plays a pivotal role in determining micellar structure and function. This work is well written, the introduction is exhaustive, the methodologies are clearly presented and appropriate and the obtained results are very interesting.

In my opinion this work can be published on the Royal Society Open Science with no modifications.

My best regards

Review form: Reviewer 2

Is the manuscript scientifically sound in its present form?

Yes

Are the interpretations and conclusions justified by the results?

Yes

Is the language acceptable?

Yes

Do you have any ethical concerns with this paper?

No

Have you any concerns about statistical analyses in this paper?

No

Recommendation?

Accept with minor revision (please list in comments)

Comments to the Author(s)

The manuscript is interesting and well-written, I have only some minor suggestions.

1. In the explanation of Figure 3, threonines in GMP are discussed only marginally although this amino acid has the highest abundance of all. I suggest to include a few words on the importance of this.
2. Also in Figure 3., I suggest to indicate the average frequency of the amino acids in mammalian proteomes (as it was mentioned for leucine). It might give further insight to the differences in the occurrences.

3. Cysteines are not discussed in this part of the paper either and although later their role is discussed in detail, I would include at least a sentence mentioning them when explaining Figure 3.
4. The vertical line representing the cleavage site between PKC and GMP is not very easy to distinguish, I suggest making it more visible, or just representing the two protein regions with horizontal bars along the X axis.

Decision letter (RSOS-190939.R0)

21-Aug-2019

Dear Dr Shields

On behalf of the Editors, I am pleased to inform you that your Manuscript RSOS-190939 entitled "Implications of kappa-casein evolutionary diversity for the self-assembly and aggregation of casein micelles" has been accepted for publication in Royal Society Open Science subject to minor revision in accordance with the referee suggestions. Please find the referees' comments at the end of this email.

The reviewers and handling editors have recommended publication, but also suggest some minor revisions to your manuscript. Therefore, I invite you to respond to the comments and revise your manuscript.

- Ethics statement

- Data accessibility

If you wish to submit your supporting data or code to Dryad (<http://datadryad.org/>), or modify your current submission to dryad, please use the following link:
<http://datadryad.org/submit?journalID=RSOS&manu=RSOS-190939>

- Competing interests

- Authors' contributions

All submissions, other than those with a single author, must include an Authors' Contributions section which individually lists the specific contribution of each author. The list of Authors

should meet all of the following criteria; 1) substantial contributions to conception and design, or acquisition of data, or analysis and interpretation of data; 2) drafting the article or revising it critically for important intellectual content; and 3) final approval of the version to be published.

- Acknowledgements

- Funding statement

Because the schedule for publication is very tight, it is a condition of publication that you submit the revised version of your manuscript before 30-Aug-2019. Please note that the revision deadline will expire at 00.00am on this date. If you do not think you will be able to meet this date please let me know immediately.

To revise your manuscript, log into <https://mc.manuscriptcentral.com//rsos> and enter your Author Centre, where you will find your manuscript title listed under "Manuscripts with Decisions". Under "Actions," click on "Create a Revision." You will be unable to make your revisions on the originally submitted version of the manuscript. Instead, revise your manuscript and upload a new version through your Author Centre.

- 1) A text file of the manuscript (tex, txt, rtf, docx or doc), references, tables (including captions) and figure captions. Do not upload a PDF as your "Main Document";

- 2) A separate electronic file of each figure (EPS or print-quality PDF preferred (either format should be produced directly from original creation package), or original software format);
- 3) Included a 100 word media summary of your paper when requested at submission. Please ensure you have entered correct contact details (email, institution and telephone) in your user account;
- 4) Included the raw data to support the claims made in your paper. You can either include your data as electronic supplementary material or upload to a repository and include the relevant doi within your manuscript. Make sure it is clear in your data accessibility statement how the data can be accessed;
- 5) All supplementary materials accompanying an accepted article will be treated as in their final form. Note that the Royal Society will neither edit nor typeset supplementary material and it will be hosted as provided. Please ensure that the supplementary material includes the paper details where possible (authors, article title, journal name).

on behalf of Dr Shaked Ashkenazi (Associate Editor) and Steve Brown (Subject Editor)
openscience@royalsociety.org

Reviewer comments to Author:
Reviewer: 1

Manuscript Number: RSOS-190939
Title: Implications of kappa-casein evolutionary diversity for the self-assembly and aggregation of casein micelles

Article Type: Research Article

Dear Editor and Authors

I checked the manuscript: RSOS-190939, entitled "Implications of kappa-casein evolutionary diversity for the self-assembly and aggregation of casein micelles" Manguy and Shields. In this work, the evolution of kappa-caseins has been systematically study, since it plays a pivotal role in determining micellar structure and function. This work is well written, the introduction is exhaustive, the methodologies are clearly presented and appropriate and the obtained results are very interesting.

In my opinion this work can be published on the Royal Society Open Science with no modifications.

My best regards

Reviewer: 2

Comments to the Author(s)

The manuscript is interesting and well-written, I have only some minor suggestions.

1. In the explanation of Figure 3, threonines in GMP are discussed only marginally although this amino acid has the highest abundance of all. I suggest to include a few words on the importance of this.
2. Also in Figure 3., I suggest to indicate the average frequency of the amino acids in mammalian proteomes (as it was mentioned for leucine). It might give further insight to the differences in the occurrences.
3. Cysteines are not discussed in this part of the paper either and although later their role is discussed in detail, I would include at least a sentence mentioning them when explaining Figure 3.
4. The vertical line representing the cleavage site between PKC and GMP is not very easy to distinguish, I suggest making it more visible, or just representing the two protein regions with horizontal bars along the X axis.

Author's Response to Decision Letter for (RSOS-190939.R0)

See Appendix A.

Decision letter (RSOS-190939.R1)

24-Sep-2019

Dear Dr Shields,

I am pleased to inform you that your manuscript entitled "Implications of kappa-casein evolutionary diversity for the self-assembly and aggregation of casein micelles" is now accepted for publication in Royal Society Open Science.

on behalf of Dr Shaked Ashkenazi (Associate Editor) and Steve Brown (Subject Editor)
openscience@royalsociety.org

Follow Royal Society Publishing on Twitter: [@RSocPublishing](https://twitter.com/RSocPublishing)
Follow Royal Society Publishing on Facebook:
<https://www.facebook.com/RoyalSocietyPublishing.FanPage/>
Read Royal Society Publishing's blog: <https://blogs.royalsociety.org/publishing/>

Appendix A

Dear Dr Ashkenazi

Please find the revised manuscript of our manuscript “Implications of kappa-casein evolutionary diversity for the self-assembly and aggregation of casein micelles” to be considered for publication in Royal Society Open Science.

We greatly appreciate the reviews that we have received. These recommendations of reviewer 2 have significantly improved our manuscript. We have made all the recommended additions (reviewers comments below in italics). The text changes in response to the reviewer are shown below in bold, and displayed in bold in the new pdf version.

REVIEWER 1 COMMENTS: no changes proposed.

REVIEWER 2 COMMENTS:

1. In the explanation of Figure 3, threonines in GMP are discussed only marginally although this amino acid has the highest abundance of all. I suggest to include a few words on the importance of this.

Page 12 paragraph 2, new second sentence: "**Threonine can be O-glycosylated and has the highest abundance in many species (supplementary figure S4)**"

2. Also in Figure 3., I suggest to indicate the average frequency of the amino acids in mammalian proteomes (as it was mentioned for leucine). It might give further insight to the differences in the occurrences.

In Figure 3, we added black lozenges to represent the average frequency of each amino acid in Swissprot, and altered the figure legend.

3. Cysteines are not discussed in this part of the paper either and although later their role is discussed in detail, I would include at least a sentence mentioning them when explaining Figure 3.

We added a new sentence to the last paragraph of page 13.

“Cysteine (C) is found exclusively in the PKC, but in different numbers in different species (see below for a discussion about possible disulphide bond gains and losses).”

4. The vertical line representing the cleavage site between PKC and GMP is not very easy to distinguish, I suggest making it more visible, or just representing the two protein regions with horizontal bars along the X axis.

In Figures 4 and S1, we now use orange (PKC) and blue (GMP) lines below each plot to instead of a thin black line to separate both parts, and changed the figure legend accordingly.

We also made a number of minor improvements to make the paper clearer.

section	change	reason
Data accessibility	changed Zenodo DOI to 10.5281/zenodo.2587122	the previous DOI was version specific, this one will always point to the latest version
Author contributions	Added "All authors have read and approved this manuscript."	Better end statement
All figures	Minor improvements to the legend	improve the figures
Figure 5	added common species names in addition of the binomial names	improve the figures
Figure S3	added common species names in addition of the binomial names	improve the figures

Thank you for considering our revisions and we look forward to your response.

Yours sincerely,

Denis Shields